# Latent CMV infection of Lymphatic endothelial cells is sufficient to drive CD8 T cell memory inflation

**Michael W. Munks**[1], **Katherine Rott**[1], **Pavlo A. Nesterenko**[1], **Savannah M. Smart**[1], **Venasha Williams**[1], **Angela Tatum**[1], **Guangwu Xu**[1], **Tameka Smith**[1], **Susan E. Murray**[2], **Ann B. Hill**[1]*

**1** Department of Molecular Microbiology and Immunology, Oregon Health and Science University, Portland, Oregon, United States of America, **2** University of Portland, Department of Biology, Portland, Oregon, United States of America

* annhillhere@me.com

**Data Availability Statement:** All relevant data are within the paper and its Supporting Information files.

## Abstract

CMV, a ubiquitous herpesvirus, elicits an extraordinarily large T cell response that is sustained or increases over time, a phenomenon termed 'memory inflation.' Remarkably, even latent, non-productive infection can drive memory inflation. Despite intense research on this phenomenon, the infected cell type(s) involved are unknown. To identify the responsible cell type(s), we designed a *Cre-lox* murine CMV (MCMV) system, where a spread-deficient (ΔgL) virus expresses recombinant SIINFEKL only in Cre+ host cells. We found that latent infection of endothelial cells (ECs), but not dendritic cells (DCs) or hepatocytes, was sufficient to drive CD8 T cell memory inflation. Infection of Lyve-1-Cre and Prox1-CreER^T2 mice revealed that amongst EC subsets, infection of lymphatic ECs was sufficient. Genetic ablation of β2m on lymphatic ECs did not prevent inflation, suggesting another unidentified cell type can also present antigen to CD8 T cells during latency. This novel system definitively shows that antigen presentation by lymphatic ECs drives robust CD8 T cell memory inflation.

## Author summary

Cytomegalovirus (CMV) establishes lifelong infection accompanied by highly unusual, very large CD8 T cell immune responses, known as "memory inflation". Memory inflation occurs even when the virus is unable to spread from one infected cell to another. In this study we used a spread-deficient virus that was genetically manipulated to allow the model protein ovalbumin (ova) expression only in host cells expressing the Cre recombinase enzyme. We used this virus to infect mice which express Cre only in specific cell types, and monitored the CD8 T cell response to ova for ~6 months. Large sustained or increasing CD8 T cell responses would indicate that the Cre-expressing cells are capable of harboring latent virus, expressing viral proteins and continuing to stimulate CD8 T cells to drive memory inflation. We show here that viral protein expression restricted to lymphatic endothelial cells is sufficient to drive robust memory inflation. However, our

**Funding:** This work was supported by NIH RO1 AI47026 and Oregon Health and Science University Presidential Bridge funding grants to ABH. The funders had no role in study design, data collection and analysis, decision to publish, or preparation of the manuscript.

**Competing interests:** The authors have declared that no competing interests exist.

results do not exclude the possibility that other cell types that might also have this capability.

## Introduction

CMVs are highly species-specific beta-herpesviruses, which establish a broadly similar host-virus relationship in all studied species. Infection typically occurs early in life and establishes a lifelong but asymptomatic latent/persistent infection. Although asymptomatic, CMV carriage has a profound impact on the host's immune system. The CD4 and CD8 T cells and NK cell compartments are expanded and distorted by cells responsive to CMV. In fact, CMV carriage was singled out as a major determinant of inter-individual variation in over 100 measurable elements of the human immune system [1]

CMV's impact on the host immune system is particularly prominent in the CD8 T cell compartment. Its two hallmarks are the enormous numbers and characteristic terminally differentiated (effector memory, short-lived effector) phenotype of CMV-specific CD8 T cells. The response is stable or increases over the individual's lifetime and is known as 'memory inflation' [2,3]. In most chronic virus infections, responses tend to be low either because viral control is good and antigen is limited, or because control is poor and high antigen load leads to T cell exhaustion. Traditionally, therefore, CMV was thought to provide just the right amount of antigen and inflammation to drive a maximal response [4,5]. However, the characteristic response to CMV occurs across the entire spectrum of virus activity–from the high virus load experienced by organ transplant patients [6], to undetectable virus following infection with a spread-deficient virus [7]. It seems likely, therefore, that some peculiar features of CMV's immunogenicity during chronic infection are responsible for the characteristic T cell response.

Many reviews have summarized our current understanding of the etiology of memory inflation in CMV infection [8–15], noting that: (i) inflation can be sustained by latent infection: viral genetic activity is required, but productive infection is not (ii) although cross-presenting DCs are needed to prime the initial T cell response, subsequent memory inflation is entirely driven by non-hematopoietic cells. However, inflation does require repeated expose to cognate antigen [16]. It is interesting that the CMV immune evasion genes that abrogate MHCI antigen presentation during lytic infection do not impact memory inflation [17], likely because these genes are not expressed by latently infected cells [18]. One model proposes that T cell sensing of reactivation leads to silencing of viral activity, followed by a strong T-cell proliferative burst [19]. The entire scenario can be driven by minute amounts of latent virus [7].

There are discordant data in the literature about the anatomical compartment in which T cells recognize antigen and proliferate to maintain the inflationary population. Torti et al. followed TCR transgenic inflationary T cells in various organs over the course of infection, and found that after four months of infection, proliferating cells could only be found within lymph nodes [20]. This provided strong evidence that $T_{CM}$ cells detect antigen within lymph nodes and differentiate into the inflationary $T_{EM}$ cells which dominate the blood. In contrast, Smith et al. used the drug FTY720 to prevent lymphocyte egress from lymph nodes. Over a five week period, naïve lymphocytes virtually disappeared from the blood, but the inflationary T cell population in the blood was not diminished and cells continued to divide. They concluded that inflationary T cells recognize antigen and are maintained entirely within an extra-nodal blood-exposed compartment [21]. The data from the first study suggest that stromal cells in the lymph node drive T cell inflation, while the data from the second study suggest that blood-exposed stromal cells of the circulatory system are responsible.

In addition to this uncertainty about the anatomical compartment(s) in which inflation occurs, the infected cell type(s) that drive inflation are not known. Although memory inflation in wild type infection could be driven at least in part by intermittent productive infection, the fact that latent CMV drives extremely robust memory inflation has led us to restrict our investigation in the current study to latently infected cells as drivers of inflation. CMV establishes latency in most organs of the body, but the scarcity of viral genomes during latency has made definitive identification of latently infected cells challenging. In murine CMV, long term latency apparently occurs only in non-hematopoietic cells [22,23]. To date, the only cell types that have been definitively shown to harbor latent virus are the liver sinusoidal EC [24] and CD31+CD146+ ECs in the lung [18]. Without definitive evidence, ECs more generally are frequently discussed as likely to be involved in driving memory inflation.

To identify cells that can drive memory inflation, we generated a virus in which the immunodominant SIINFEKL epitope from ovalbumin is encoded behind a lox-stop-lox (LSL) sequence, and thus expressed only after Cre-mediated excision of the stop sequence. SIINFEKL expressed behind an immediate-early *(IE)* MCMV promoter drives a very large inflationary response which suppresses responses to other epitopes [25], making it an ideal single epitope reporter with which to interrogate the inflationary response. If the virus were to spread to a new cell type after *LSL* excision, this would confound our interpretation. Therefore, the virus was rendered spread-deficient to ensure that antigen remained restricted to the Cre-expressing cell. Using this virus (ΔgL-LSL-SL8) to infect Cre-expressing mice, we demonstrate that latent infection of lymphatic ECs is sufficient to drive CD8 T cell memory inflation in response to MCMV infection.

## Results

### Cell-specific antigen expression can be used to interrogate the process of memory inflation

Following infection of Cre-transgenic mice, cellular Cre has been shown to efficiently excise the STOP sequence from MCMV-encoded *lox-stop-lox* (*LSL*) constructs, in which a transcriptional and translational STOP sequence is flanked by *loxP* sites [26]. To create a virus where SIINFEKL (SL8) expression could be limited to specific cell types, an *LSL* sequence was cloned in front of *GFP-SL8* within the *IE2* locus. To ensure that only Cre-expressing cells would present SIINFEKL, it was essential to prevent progeny virions with the stop sequence excised from going on to infect other cells. To this end, we also disrupted the gene encoding glycoprotein L (gL), which is essential for viral entry into cells. The resultant virus, ΔgL-LSL-SL8 (**Fig 1A**), was propagated on complementing cells. This virus can enter cells and undergo fully lytic replication or establish latency. However, all resulting progeny virions lack gL and are incapable of infecting secondary cells.

We investigated the integrity of the *LSL* sequence by infecting cells *in vitro* with ΔgL-LSL-SL8. Because the SIINFEKL epitope was linked to GFP, we analyzed GFP expression by FACS. Unexpectedly, a very small proportion (0.033%) of ΔgL-LSL-SL8-infected cells expressed GFP, albeit with a low intensity, whereas most cells infected with ΔgL-SL8, which lacks the *LSL*, had robust GFP expression (**Figs 1B, 1C and S1A**). Using primers that could distinguish an intact *LSL* from a recombined *LSL* (**S1B Fig**), we performed qPCR on a new clone of the bacterial artificial chromosome (BAC) DNA from which the ΔgL-LSL-SL8 virus was generated, on DNA isolated from a viral stock, and on cDNA generated from infected cells. In each case, we found a similar, very low level of spontaneous *LSL* recombination (**Fig 1D**). We concluded that spontaneous *loxP* recombination occurs very infrequently in the BAC DNA during propagation, which may cause the small amount of GFP expression observed in

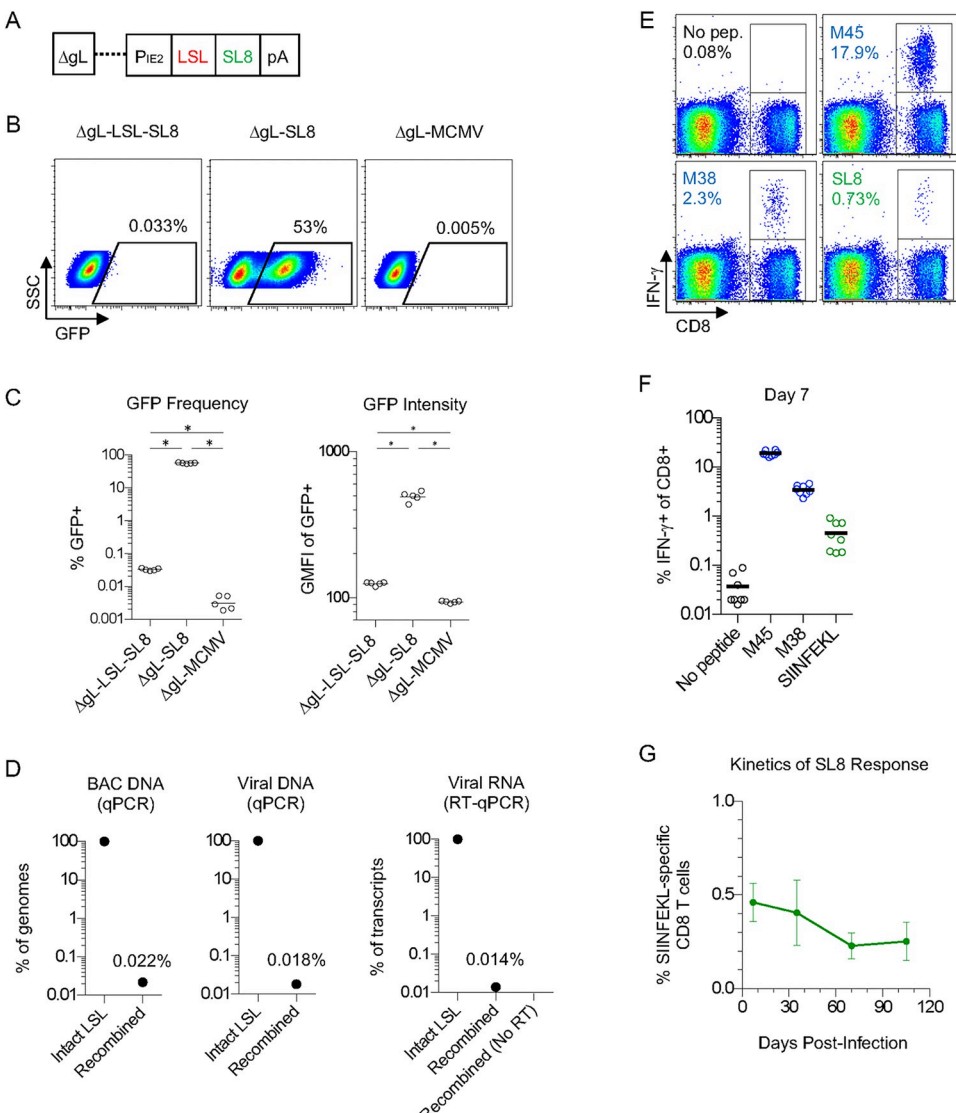

**Fig 1. ΔgL-LSL-SL8 MCMV Infection Causes CD8 T Cell Priming but Not Inflation.** (A) Diagram of ΔgL-LSL-SL8 MCMV. (B and C) NIH 3T3 cells, infected at 1 MOI overnight with the indicated virus, were analyzed by FACS for GFP expression. Representative FACS plots (B) and all 5 biological replicates (C) are shown. (* p < 0.001) (D) DNA from the ΔgL-LSL-SL8 BAC, from which virus was derived (left), and DNA from viral stock (middle), was analyzed by qPCR to quantitate the frequency of DNA with intact LSL sequence vs. recombined LSL. (right) gL-3T3 complementing cells were infected at 1 MOI overnight with ΔgL-LSL-SL8. RNA was reverse-transcribed to cDNA. The cDNA was used for qPCR in parallel with the BAC DNA and viral DNA samples to allow direct comparison. (E, F and G) Mice were infected with ΔgL-LSL-SL8 (n = 8). After 7 days, blood lymphocytes were stimulated with the indicated peptides and analyzed by ICS for IFN-γ production. The frequency of CD8 T cells that were IFN-γ+ are shown for a representative mouse (E) and for all mice (F). Peripheral blood CD8 T cells were analyzed longitudinally for SIINFEKL-specific CD8 T cells by ICS for IFN-γ (G). All data are representative of two or more independent experiments. Mean +/- S.E.M. is graphed. Statistical significance was determined by Student's t test for paired data.

Cre-negative infected cells (Fig 1B). However, it is also possible that a low level of transcriptional and/or translational readthrough of the stop sequence occurs during lytic viral infection

We next investigated the impact of this low level of Cre-independent GFP-SL8 expression on the T cell response to the virus *in vivo*. In order for our system to work as intended, it was critical that the ΔgL-LSL-SL8 virus would not drive memory inflation in the absence of Cre.

However, it would be advantageous if it could prime SIINFEKL-specific CD8 T cells in the absence of Cre, in order to remove suboptimal priming as a trivial explanation for lack of memory inflation. We therefore infected C57BL/6 mice with ΔgL-LSL-SL8, and monitored the SIINFEKL-specific response in the peripheral blood over time. A small SIINFEKL-specific response, comprising ~0.5% of CD8 T cells, was evident at day 7 (**Fig 1E and 1F**). Critically, however, this response did not undergo inflation during latent infection (**Fig 1G**). We concluded that Cre-independent expression of SIINFEKL during acute infection with ΔgL-LSL-SL8 primed a population of SIINFEKL-specific T cells, but did not cause memory inflation. We further concluded that infection with ΔgL-LSL-SL8 could be used, without additional experimental manipulation, to interrogate the ability of cell-specific Cre-driven SIIN-FEKL expression to drive memory T cell inflation.

## Inflation is driven by Latent Virus in ECs

Hepatocytes and ECs are major targets of infection in the first 24 hours, and DCs are infected within 48 hours [26,27]. We infected CD11c-Cre mice (targeting DCs), albumin-Cre mice (targeting hepatocytes) and Tek-Cre mice (targeting ECs) with ΔgL-LSL-SL8 to determine which of these cell types, if any, can drive T cell memory inflation. At day 7, each mouse strain developed a strong response to the endogenous MCMV M45 epitope, a non-inflationary epitope (**Fig 2A**). This indicated that all strains were adequately infected, and mounted a similarly strong CD8 T cell response. At this time point, SIINFEKL-specific responses constituted ~0.5% of CD8 T cells in Cre-negative wildtype C57BL/6 mice and in each of the Cre+ mouse strains (**Fig 2B**). However, the subsequent behavior of primed populations was markedly different. In C57BL/6 mice, CD11c-Cre mice, and albumin-Cre mice, the SIINFEKL-specific response decreased and was maintained at a low level, displaying the classic kinetics of a central-memory CD8 T cell response that occurs when antigen is no longer detected. In contrast, the SIINFEKL-specific response in Tek-Cre mice underwent classic memory inflation, increasing to an average of 1.4% of CD8 T cells by 3 months. The experiment provides strong evidence that antigen expressed by latently infected ECs is able to drive CD8 T cell memory inflation.

## Inflation is driven by Lymphatic ECs

The data shown in Fig 2 demonstrated that inflation can be driven by latently infected ECs. ECs line blood vessels and lymphatic vessels, as well as some specialized compartments

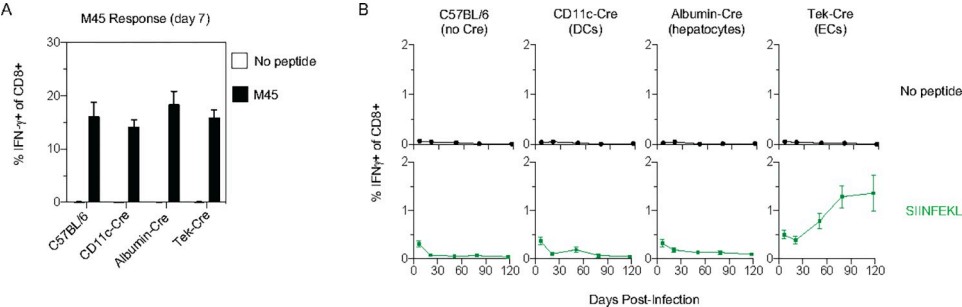

**Fig 2. Antigen from Latently-Infected ECs is Sufficient to Drive CD8 T Cell Memory Inflation. (A and B)** C57BL/6, CD11c-Cre, Albumin-Cre and Tek-Cre mice were infected with ΔgL-LSL-SL8 on day 0. **(A)** On day 7, peripheral blood was stimulated with M45 peptide, and CD8 T cell production of IFN-γ was analyzed by ICS, to confirm equivalent infection between strains. **(B)** In the same mice, peripheral blood CD8 T cells were analyzed longitudinally by ICS for IFN-γ after stimulation with SIINFEKL, or vehicle alone. Data are representative of two independent experiments with 6–8 mice per group. Mean +/- S.E.M. is graphed.

contiguous with the circulatory system. We were interested in determining whether we could identify subtypes of ECs that are capable of driving memory inflation. MCMV establishes latency in many organs of the body, including the liver, lung, spleen and kidneys. In a meticulous study of latency in the liver, latent virus was found in liver sinusoidal ECs, but no latent virus was found in cells of hematopoietic origin, hepatocytes, or other EC types [24]. Liver sinusoidal ECs share some functional and phenotypic qualities and functions with lymphatic ECs, including expression of Lyve-1 [28]. For this reason, we infected Lyve-1-Cre mice with ΔgL-LSL-SL8, enabling SIINFEKL expression only in Lyve-1-expressing cells. Lyve-1-Cre mice, like C57BL/6 and albumin-Cre controls, developed a robust CD8 T cell response to M45 (**Fig 3A**). All strains developed an SIINFEKL-specific response at day 7, but only Lyve-1-Cre mice developed a strong SIINFEKL-specific inflationary responses (**Fig 3B**).

Lyve-1 is expressed both on lymphatic ECs and liver sinusoidal ECs. As described above, one prior study concluded that inflation occurs primarily in the lymph nodes [20], more consistent with lymphatic ECs driving inflation, while another study concluded that inflation occurs primarily in the blood [21], more consistent with liver sinusoidal ECs driving inflation. To distinguish between these competing models, we used Prox1-CreER$^{T2}$ mice [29]. Prox1 is expressed on lymphatic ECs but not on liver sinusoidal ECs [28]. It is also expressed on hepatocytes [30,31]. However, because hepatocytes are not known to harbor latent virus [24], and inflation did not occur in albumin-Cre mice (Fig 2B), we excluded hepatocytes as a driver of inflation.

Prox1-CreER$^{T2}$ mice were treated with tamoxifen to induce expression of the Cre transgene, then infected with ΔgL-LSL-SL8. Tek-Cre+ and Tek-Cre- mice, also infected with ΔgL-LSL-SL8, were used as positive and negative controls for SIINFEKL inflation. CD8 T cells specific for the conventional endogenous M45 epitope, or specific for the inflationary endogenous M38 epitope, had similar kinetic patterns in all three mouse strains (**Fig 3C**). This indicates that tamoxifen's estrogenic effects would not complicate our interpretation of the SIINFEKL-specific response. SIINFEKL-specific CD8 T cells inflated in Tek-Cre+ mice, as expected, and also in tamoxifen-treated Prox1-CreER$^{T2}$ mice (**Fig 3C**). This demonstrated that latently-infected lymphatic ECs are capable of driving memory inflation.

## Antigen presentation by Lymphatic ECs is sufficient but not necessary for memory inflation

While these data indicate that EC presentation of antigen is sufficient to drive memory inflation, they do not exclude a role for other cell types. To investigate whether ECs are uniquely capable of driving inflation, we set up a system where all cells are capable of presenting SIINFEKL, except for ECs. We again used a Cre-lox system, but this time to conditionally delete beta-2-microglobulin (β2m), the MHC I light chain. We bred Tek-Cre+;β2m$^{fl/fl}$ mice, so that ECs would lack MHC I. Hematopoietic stem cells arise from a progenitor that also gives rise to ECs during development [32], and thus transiently express Tek-Cre. Because some Tek-Cre+; β2m$^{fl/fl}$ mice lacked MHC I on hematopoietic cells, we reconstituted all mice with WT bone marrow, ensuring that only ECs lacked the ability to present viral peptides. We found that deletion was efficient in lymphatic ECs, with ~80% of these cells lacking MHC I expression (**Fig 4A–4C**). Meanwhile, MHC I expression on blood ECs was significantly downregulated on the entire population (**Fig 4B**), but completely absent on only a small fraction of cells. This could indicate either incomplete penetrance (i.e. incomplete floxed β2m excision), or that β2m, which is present in significant quantities in serum, was acquired by blood ECs and used to stabilize MHC I surface expression.

To enable Cre-negative cells to present SIINFEKL, we used a different MCMV strain, ΔgL-SL8, which is spread-deficient but does not depend on Cre for SIINFEKL expression

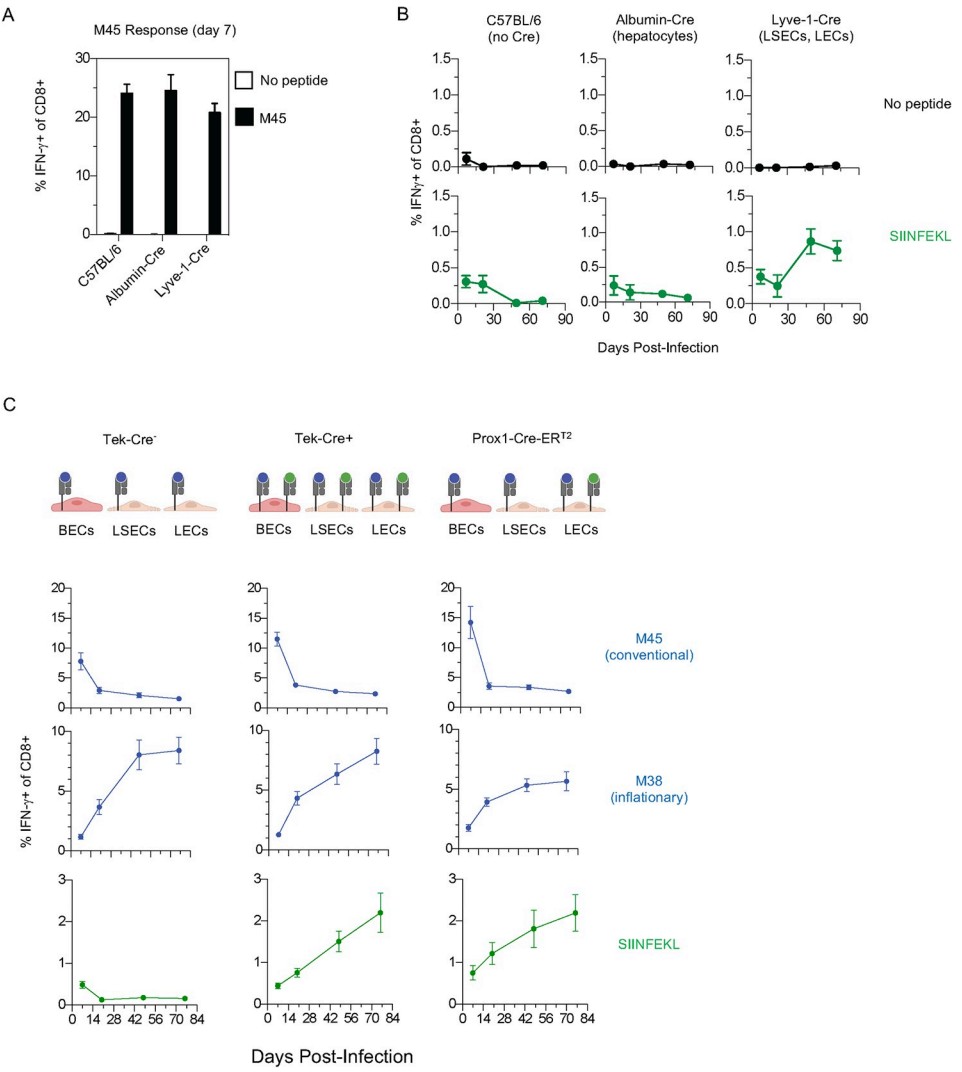

**Fig 3. Lymphatic ECs (LECs) are Sufficient to Drive CD8 T Cell Inflation. (A and B)** C57BL/6, Albumin-Cre and Lyve-1-Cre mice were infected with ΔgL-LSL-SL8 on day 0. **(A)** On day 7, peripheral blood was stimulated with M45 peptide, and CD8 T cell production of IFN-γ was analyzed by ICS, to confirm equivalent infection between strains. **(B)** In the same mice, peripheral blood CD8 T cells were analyzed longitudinally by ICS for IFN-γ after stimulation with SIINFEKL, or vehicle alone. Data are representative of two independent experiments with 6–8 mice per group. **(C)** The indicated mice were infected with ΔgL-LSL-SL8 on day 0. Prox1-CreER[T2] mice were injected with 1 mg of tamoxifen daily, from day -1 to day 7. Peripheral blood CD8 T cells were analyzed longitudinally for epitope-specific CD8 T cells by ICS for IFN-γ. Data are representative of two independent experiments with 4 to 7 mice per group. Mean +/- S.E.M. is graphed.

(**S1A Fig**). Following infection, CD8 T cells underwent inflation in Tek-Cre[+];β2m[fl/fl] mice that was equivalent, if not greater than in Tek-Cre[-] littermate controls, as well as C57BL/6 mice (**Fig 4D**). These data suggest that cells other than lymphatic ECs are capable of driving inflation.

## Discussion

The data described above shows that latent infection of ECs, but not DCs or hepatocytes, is sufficient to drive CD8 T cell memory inflation to an MHC I epitope expressed by those cells. In

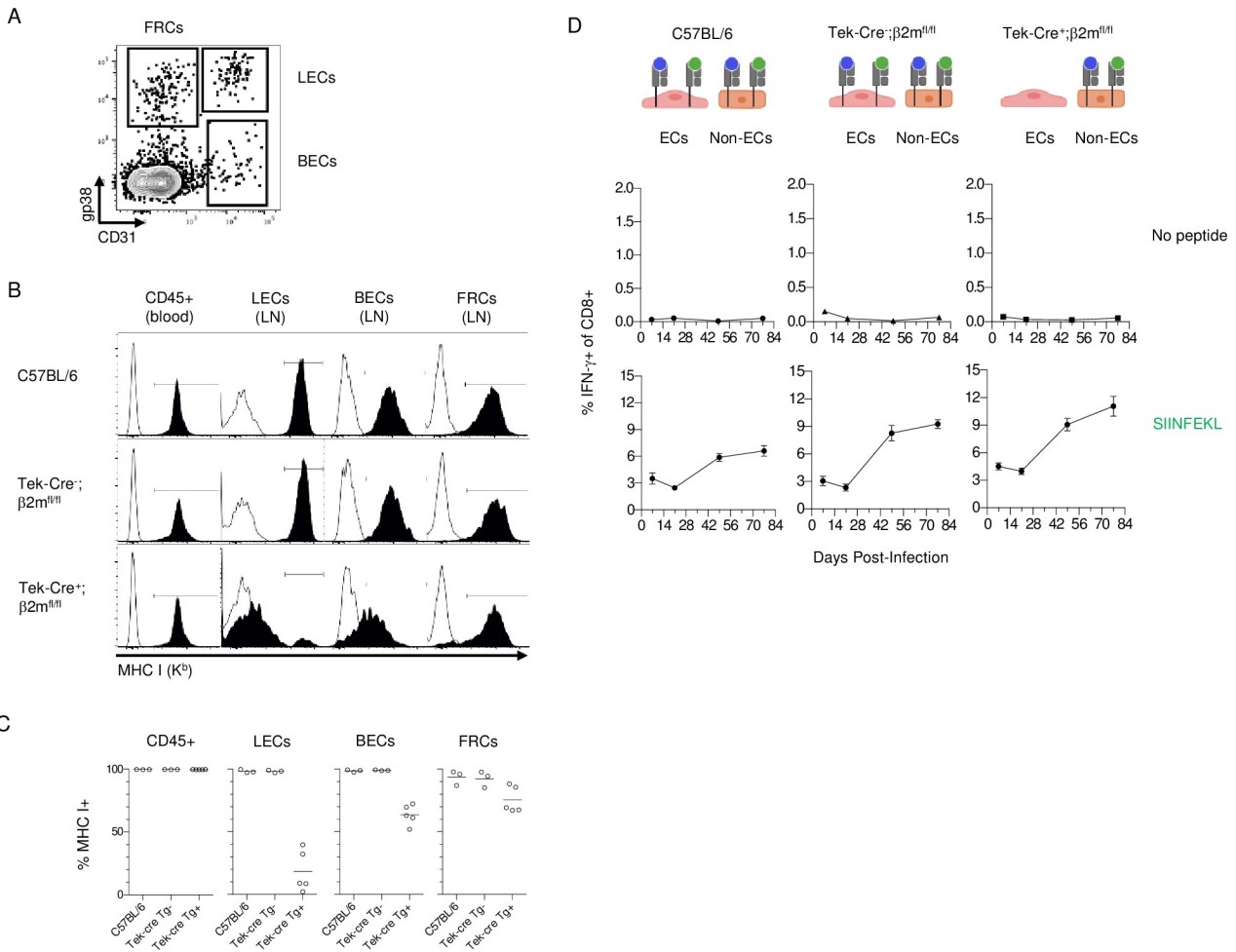

**Fig 4. Antigen Presentation by Lymphatic ECs is not Required for CD8 T Cell Inflation.** C57BL/6, Tek-Cre⁻;β2m^fl/fl littermates and Tek-Cre⁺; β2m^fl/fl mice were infected with ΔgL-SL8 MCMV. **(A)** At the conclusion of the experiment, lymph nodes stromal cells (CD45⁻) were identified as lymphatic ECs (LECs; CD31⁺gp38⁺), blood ECs (BECs; CD31⁺gp38⁻), or fibroblastic reticular cells (FRC; CD31⁻gp38⁺). **(B and C)** MHC I expression (solid histograms) was compared to the fluorescence minus one (FMO) staining (open histograms) of the same cell population. One individual mouse is shown in **(B)**, and all mice in the group are shown in **(C)**. **(D)** CD8 T cell responses to SIINFEKL were analyzed in the blood longitudinally by ICS for IFN-γ. Graph shows mean +/- S.E.M.

particular, we found that antigen expression by lymphatic ECs was sufficient to induce a robust inflationary response.

However, when β2m was not expressed by ECs, inflation still occurred. While this strongly suggested that cells other than lymphatic ECs are capable of driving inflation, we cannot make that conclusion definitively. MHC class I heavy chain can bind peptide in the absence of β2m, exit the ER, and form a stable complex at the cell surface with β2m [33]. Since β2m is abundant in serum, we hypothesize that this process led to the intermediate MHC I expression we observed on blood ECs in Tek-Cre⁺;β2m^fl/fl mice (Fig 4B). A high affinity peptide such as SIINFEKL is particularly efficient at stabilizing heavy chain, and could be present at the cell surface even when MHC I is below the threshold of detection by FACS. Since as few as 10 peptide-MHC complexes are needed to activate CD8 T cells [34], it remains possible that sufficient SIINFEKL could have been presented on the surface of lymphatic ECs to account for the memory inflation we observed even in mice whose lymphatic ECs genetically lacked β2m. An additional complication could occur if some ECs were generated from the B2m+ donor marrow, as

observed in the liver by Seckert et al. [24], even though those authors did not find latent virus in the donor-derived ECs. Altogether, we believe that the low MHC1 on most ECs in Fig 4 means that the most likely interpretation of this experiment is that cells other than ECs can drive inflation. However, further experiments will be needed to solidify this conclusion.

In this paper we provide definitive evidence that antigen presented on latently infected lymphatic ECs is sufficient to drive memory inflation. In other settings, lymphatic ECs play a role in peripheral CD8 T cell tolerance and deletion [35–37]. It was therefore surprising that during chronic MCMV infection, they elicited expansion of inflationary T cells so robustly. Furthermore, this occurred during latent infection, with a spread-deficient virus, a context where innate inflammation was presumably minimal or absent. We hypothesize that CMV-specific CD4 T cells may be responsible for rendering these normally tolerogenic cells highly immunogenic, in a process analogous to CD4 T cell licensing of immature DCs. CD4 T cells are known to be important for CD8 T cell memory inflation [38,39]. These CD4 T cells could induce co-stimulatory molecules, such as 4-1BB, OX-40 and CD70, all of which promote inflation [40–42]. Lymphatic ECs do not express these co-stimulatory molecules at steady state. However, lymphatic ECs highly express MHC II, so MCMV-specific CD4 T cells may act on latently-infected lymphatic ECs, turning these normally tolerogenic cells into potent stimulators of CD8 T cell proliferation.

The enormous T cell response to CMV has appropriately been called "infamous," and "perhaps the strongest modulator of the T cell compartment known" [43]. Here we have shown that this massive modulation can be sustained even when antigen presentation by CMV is limited to latent infection of lymphatic ECs.

## Materials and methods

### Ethics statement

All studies were approved by the Institutional Biosafety Committee and the Institutional Animal Care and Use Committee at Oregon Health and Sciences University.

### Virus strains and infections

Mice were infected i.v. with $2 \times 10^5$ PFU of virus. Wildtype MCMV was of the strain MW97.01 [44], which was generate from the Smith-derived bacterial artificial chromosome (BAC) pSm3fr [45]. The viruses used in this paper were generated from pSm3fr. MCMV-EGFP-SL8 expresses EGFP tagged at the C terminus with SIINFEKL plus 7 N-terminal amino acids from ovalbumin (SGLEQLESIINFEKL), to facilitate normal peptide excision, behind the *IE2* promoter [25]. MCMV-LSL-SL8 was generated by inserting an *LSL* sequence into the EGFP-SL8 plasmid. This fusion construct was then targeted to replace the *IE2/m128* gene using established techniques [46].

To produce the ΔgL viruses ΔgL-SL8 and ΔgL-LSL-SL8, an ampicillin gene fragment was inserted into the *gL* gene of the MCMV-EGFP-SL8 and MCMV-LSL-EGFP-SL8 BACs respectively, using homologous recombination. After recombination, an individual clone of the virus was selected that had lost the BAC sequences. Stocks of these viruses were generated in gL-3T3 cells, which provide gL in *trans* [7,38], and titered by plaque assay on gL-3T3s without centrifugal enhancement. Each individual ΔgL virus stock was checked for reversion by infecting BALB 3T3s, a non-complementing cell line, then passaging and monitoring these infected cells for 30 days to confirm that no further signs of infection developed as a result of cell to cell spread.

## Cell lines

gL-3T3s were the kind gift of Helen Farrell. NIH-3T3 and BALB 3T3s were obtained from ATCC.

## PCR analysis of *LSL* excision

Quantitation of intact vs. excised lox-stop-lox sequences from BAC DNA and viral DNA was performed by qPCR. Quantitation from viral mRNA was performed by RT-qPCR. BAC DNA was purified using the NucleoBond BAC 100 kit (Takara). Viral DNA was isolated from viral stocks by digestion with proteinase K (Roche) at 55° C, followed by isopropanol precipitation. To isolate viral RNA, gL 3T3 cells were infected at an MOI of 1, or left uninfected, for 16–18 hours, before RNA was isolated using TRIzol (Invitrogen). cDNA was produced using Super-Script IV VILO Master Mix (Invitrogen/ThermoFisher). Two sets of qPCR primers were designed to produce amplicon either only from intact DNA, or only from excised DNA. Titrated amounts of custom synthetic dsDNA templates (gBlocks), corresponding to intact or excised DNA, were used for quantitation. Primers and dsDNA templates were from Integrated DNA Technologies. qPCR reactions were set up using Power SYBR Green PCR Master Mix (Applied Biosystems/ThermoFisher). Ct values and melt curve analyses were performed in the OHSU Gene Profiling Shared Resource with a QuantStudio 12K Flex system (Applied Biosystems/ThermoFisher).

## Mice

The following strains were purchased from the Jackson Laboratory: C57BL/6J, Tek-Cre (B6. Cg-Tg(Tek-Cre)1Ywa/J; stock 008863; also known as Tie-2 Cre), Lyve-1-Cre (B6:129P2-Lyve1tm1.1(EGFP/Cre)Cys/J; stock 012601), Alb-Cre (B6.Cg-Speer6-ps1Tg(Alb-Cre_21Mgn)/J; stock 003574), CD11c-Cre (B6.Cg-Tg(Itgax-Cre)1-1Reiz/J; stock 008068). Prox-1-CreERT2 mice (Bazigou et al., 2011) [29] were the kind gift of Taija Makinen. β2mfl/- mice (C57BL/6N-B2mtm1c(EUCOMM)Wtsi/Tcp) were cryo-recovered from frozen sperm at the Canadian Mutant Mouse Repository, Toronto Centre for Phenogenomics, 25 Orde St, Toronto, Ontario, Canada. Heterozygous β2mfl/- mice were bred to homozygous Tek-Cre mice, then back crossed to β2mfl/- mice to generate Tek-Cre+/-,β2mfl/fl mice.

## Tamoxifen induction

Tamoxifen (Sigma) was dissolved in corn oil at 40 mg/ml and stored up to 10 days at 4 C. To induce Cre expression in Prox-1-CreERT2 mice, 1 mg of tamoxifen was injected i.p. daily, from day -1 to day 7 relative to infection.

## Intracellular cytokine staining and FACS analysis

For measurement of intracellular IFN-γ, peripheral blood was collected at the indicated time points. Red blood cells were lysed with 3 ml of lysis buffer (150 mM NH4Cl, 10 mM NaHCO3) and the remaining cells were incubated for 5–6 hrs at 37°C in the presence of 1 uM of the indicated peptide and brefeldin A (GolgiPlug; BD Pharmingen). Surface staining was done overnight at 4 degrees, and cells were fixed and permeabilized for intracellular cytokine staining with Cytofix/Cytoperm (BD Pharmingen). The following fluorescently conjugated antibodies were used (CD8α [clone 53–6.7], IFN-γ [clone XMG1.2]), and all purchased from either BD Biosciences, eBioscience, or BioLegend. Samples were acquired on an LSR II (BD) or Fortessa (BD) and analyzed with FlowJo software (Tree Star).

## Lymph node digestion and staining

Lymph nodes were digested and stained as previously described (Lane et al, 2018). Briefly, lymph node capsules were manually disrupted, then digested with 1 mg/ml collagenase IV (Worthington) plus 40 ug/ml DNase I (Roche) to release the lymphocytes. The remaining tissue was then digested with 3.3 mg/ml collagenase D plus 40 ug/ml DNase I to dissociate the stroma. The stroma was stained for CD45 (clone 30-F11), CD31 (clone MEC13.3) and gp38/podoplanin (clone 8.1.1).

## Supporting information

**S1 Fig. Viruses and qPCR Quantitation. (A)** Diagram of MCMV strains used in this study, all of which are grown on complementing gL-3T3 cells. ΔgL-LSL-SL8 MCMV has a lox-stop-lox (LSL) cassette which prevents constitutive expression of GFP with SL8 (SIINFEKL) linked to its C-terminus. GFP-SL8 is expressed after cre-dependent excision of the stop sequence. ΔgL-SL8 MCMV constitutively expresses GFP-SL8. ΔgL-MCMV lacks the gene encoding the essential glycoprotein gL. **(B)** Diagram of qPCR primer binding sites. When there is an intact LSL, primers 2 and 3 produce a ~100 bp amplicon. When there is a recombined LSL (i.e. a single loxP site), primers 1 and 3 produce a ~100 bp amplicon.
(TIF)

## Acknowledgments

We thank Amanda Lund and Chris Snyder for invaluable discussion. We thank Amanda Lund, Victor Engelhardt and Taija Makinen for the Prox1-CreER[T2] mice. We thank Helen Farrell and Nick Davis-Poynter for the gL-3T3 cells.

## Author Contributions

**Conceptualization:** Ann B. Hill.

**Data curation:** Michael W. Munks, Katherine Rott, Pavlo A. Nesterenko, Savannah M. Smart, Venasha Williams, Angela Tatum, Guangwu Xu, Tameka Smith, Ann B. Hill.

**Formal analysis:** Michael W. Munks, Pavlo A. Nesterenko, Savannah M. Smart, Ann B. Hill.

**Funding acquisition:** Ann B. Hill.

**Investigation:** Michael W. Munks, Katherine Rott, Pavlo A. Nesterenko, Savannah M. Smart, Venasha Williams, Angela Tatum, Ann B. Hill.

**Methodology:** Michael W. Munks, Angela Tatum, Guangwu Xu, Tameka Smith, Susan E. Murray, Ann B. Hill.

**Project administration:** Michael W. Munks, Ann B. Hill.

**Supervision:** Guangwu Xu.

**Writing – original draft:** Michael W. Munks, Ann B. Hill.

**Writing – review & editing:** Michael W. Munks, Susan E. Murray, Ann B. Hill.

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
