## [Decision Letter · Decision Letter 0]

29 Apr 2022

Dear Dr. Hill,

Thank you very much for submitting your manuscript "CD8 T Cell Memory Inflation is Driven by Latent CMV Infection of Lymphatic Endothelial Cells" for consideration at PLOS Pathogens. As with all papers reviewed by the journal, your manuscript was reviewed by members of the editorial board and by several independent reviewers. The reviewers appreciated the attention to an important topic. Based on the reviews, we are likely to accept this manuscript for publication, providing that you modify the manuscript according to the review recommendations.

The reviewers appreciated the quality and importance of the work and ask only that you consider modifying the text to improve clarity.

Sincerely,

Robert F. Kalejta

Associate Editor

PLOS Pathogens

Blossom Damania

Section Editor

PLOS Pathogens

Kasturi Haldar

Editor-in-Chief

PLOS Pathogens

orcid.org/0000-0001-5065-158X

Michael Malim

Editor-in-Chief

PLOS Pathogens

orcid.org/0000-0002-7699-2064

The reviewers appreciated the quality and importance of the work and ask only that you consider modifying the text to improve clarity.

Reviewer Comments (if any, and for reference):

Reviewer's Responses to Questions

**Part I - Summary**

Reviewer #1: Mike Munks and colleagues and revived Ann Hill provide here a well-planned and well-conducted study using an elegant strategy to positively identify endothelial cells (EC), specifically lymphatic EC, as drivers of CD8 T-cell expansion during latent MCMV infection, a phenomenon known as “memory inflation” (MI). While previous work, primarily from the Reddehase group, has proposed EC to be strong candidates for driving MC, based on the demonstration that latent viral genome almost quantitatively localizes to EC and that genes that code for antigenic peptides during latency can be expressed from latent viral genomes, the merit of the here presented work is to close a gap in the chain of evidence by showing directly that MI can take place when expression of a reporter antigenic peptide, specifically SIINFEKL, is released from a lox-stop-lox block by Cre-recombination selectively in EC in general but also if Cre-recombination is restricted specifically to lymphatic EC. What adds to the elegance is the use of recombinant virus combining lox-stop-lox with deletion of gL to prevent virus spread and thus to keep all events restricted to the Cre-expressing cells analysed. Nonetheless, I think the work would benefit from considering my comments. Actually, my main concern is the title of the manuscript. As far as I can see, and also confirmed by the Abstract, the data provide positive evidence to conclude that lymphatic EC are sufficient to drive MI, but I see no negative evidence that would exclude other EC subtypes, such as the liver-sinusoidal EC (LSEC), so far the only cell type (in the mouse model) molecularly identified to become latently infected, or EC of the widely-ramified capillary bed of the lungs, from being equally good drivers of MI if one would exclude SIINFEKL expression selectively in lymphoid EC. As cursory readers tend to take the message from the title without reading the manuscript, I would strongly suggest to restrict the title claim to what has been positively shown. I believe the paper would even profit from the more general statement: “CD8 T Cell Memory Inflation is driven by Latent Infection of Endothelial Cells”. As long as non-lymphatic EC are not excluded, I would leave it to the Abstract to say that lymphatic EC have here been identified as being sufficient, leaving it to the Discussion if other EC subtypes might be sufficient as well. Actually, I currently see no rationale for assuming that they would not do the job.

Reviewer #2: This work by Munks and colleagues addresses a critical area of CMV-related research: what are the important sources of latent/persistent viral genome that drive the phenomena known as T-cell memory inflation. Memory inflation does occur to some extent in other herpesviruses, but is most pronounced in the case of CMV infected vertebrates, including mice, monkeys and humans. The results here are consistent with past published data showing that, in vitro, MCMV can establish latency in liver LSECs and can be reactivated from these cells (Cicin Sain et al). This paper uses clever mouse and viral genetic tools and experimental approaches to show that endothelial cells are a key source of viral genome that drives T-cell inflation in vivo, which in this reviewers opinion significantly advances our understanding of cellular latency reservoirs that need to be considered as we attempt to decipher why this virus has such a huge impact on our circulating immune cell compartment.

Reviewer #3: I read with interest the manuscript “CD8 T Cell Memory Inflation is Driven by Latent CMV Infection of Lymphatic Endothelial Cells” by M. Munks et al. The paper addresses the relevant question on the cell type that drives memory inflation and shows that endothelial cells are sufficient but not necessary for the induction of this phenomenon. The paper is overall clear and well written, providing robust evidence to support the claim by the authors. The manuscript will be of interest to scientists in the field and well-deserving of a publication in PLOS Pathogens, provided that some minor issues are clarified:

**Part II – Major Issues: Key Experiments Required for Acceptance**

Reviewer #1: I do not ask for additional experiments.

Reviewer #2: -The Hill group has utilized the ∆gL virus as a ‘spread deficient’ model in several papers, but has never reported how long they believe that viral antigen is expressed in initially infected cells in vivo prior to this mutant going latent or being cleared. Can they provide any data on this? Have they ever tried to look at this using their ∆gL-GFP virus in vivo?

-Does FTY720 inhibit splenic egress/transit in the experimental setups and doses that the authors use? I ask this because the Spector Lab years ago showed that MCMV establishes latency in splenic marginal zone fibroblast reticular cells (FRCs) and these cells express good levels of MHC-I, and likely also MHC-II when activated based on the work of Shanon Turley and others.

Reviewer #3: None

**Part III – Minor Issues: Editorial and Data Presentation Modifications**

Reviewer #1: Specific issues:

INTRODUCTION

(1) The authors show here MI on the basis of functional, IFNg-secreting CD8 T cells. The first evidence for MI of IFNg-secreting CD62L- effector-memory cells associated with latent CMV infection was in Holtappels et. al. JVI 2000. The title said already everything on which the here presented work is based 22 years later. Karrer et al. 2003 have used multimer-staining (instead of function), a more complete time course, and named the phenomenon MI. I think it would be fair to cite both papers and not ignore the one that came first.

(2) Again regarding correct referencing, in the third paragraph the authors cite a number of reviews on MI (mostly from authors who never worked on viral latency and expression of antigenic peptides following transcription from latent viral genomes) but the review that is closed to the topic of the present work, namely the latently infected cell type and viral gene expression from latent viral genomes driving MI, is curiously ignored (Seckert et al. Med. Microbiol. Immunol. 2012).

(3) Page 4, end of first §. The statement “driven by minute amounts of latent virus” is wrong or at least misleading. One might argue that compared to productive infection genome load is generally low in latency, but this is trivial. The truth is that a high load of latent virus favors MI, simply because the number of genomes from which antigenic peptides can be expressed is high. Accordingly, MI is usually observed after high-dose intraperitoneal or intravenous infections resulting in a high load of latent virus, but poor or absent after local infections that result in a low latent genome load (Snyder et al. PLoSPathog. 2011; Holtappels et al. Vaccines, 2020). The requirement for massive primary infection to generate a high latent genome load is also the reason for the fact that- unlike always claimed- MI is not usually observed in human CMV infection (Jackson et al. Med. Microbiol. Immunol. 2019).

(4) It has long been disputed if MI is explained by immunological parameters, such as IL-15, or if antigen presentation during viral latency is the crucial driver. This was mainly because latency was long considered as a state during which the viral genes are all silenced. It may be worthwhile to mention that the need for antigen expression has been conclusively documented only recently by showing that SIINFEKL-specific MI fails in latently infected mice selectively not presenting SIINFEKL because of a point mutation L8A, although the cytokine environment of the latently infected mice supported MI for intrinsic epitopes (Lemmermann and Reddehase, Pathogens 2021). Previous work had shown that inflation does not occur in uninfected mice, an approach that was inconclusive as uninfected mice trivially do not express antigen but also lack an MI-supporting cytokine environment.

(4) Page 5, first §, line 6: although listed on the cover page, please define non-standard abbreviations also in the text upon first occurrence to aid the reader: here: LSL.

RESULTS

(1) I greatly appreciate the care given to “leakiness” of the LSL by spontaneous recombination or read-through. Notably this leakiness was sufficient to prime SIINFEKL-specific CD8 T cells but –fortunately, not sufficient for driving MI. It came into my mind that there might exist cell-type specific differences. The phenomenon was observed for NIH 3T3 in cell culture. I do not ask to test for leakiness in EC cultures and BMDC cultures, and in vivo the situation may be different anyway.

Apparently, professional APC can present antigenic peptide from LSL virus. The authors should at least consider and discuss the issue of cell-type. Leakiness-based priming of SIINFEKL-specific CD8 T cells was ca. 0,5% of all CD8 T cells —this is something, because after MI at 3 months the frequency was 1.4%, which is not miles away! Can the authors tell the pre-MI magnitude of the response to SIINFEKL with a control virus lacking LSL?

(2) p. 6 last paragraph. The work by Sacher in Host Cell & Microbe, 2008 should be cited in addition to Hsu.

(3) When discussing latency in EC, the authors should also cite the recent work by Griessl et al. ( Front. Immunol. 2021) who have shown, although buried as a Suppl. Fig., that latent viral DNA in the lungs localizes to CD31+CD146+ EC, which adds a second EC subtype to the LSEC described before.

Do the authors have direct evidence for latent viral DNA localizing to lymphatic EC- in addition to the indirect evidence from MI?

(4) The strategy to restrict SIINFEKL expression to lymphatic EC is most elegant. Although I do not distrust the work by Mouta Carreira, 2001, I have learnt that EC biology is a difficult and mined field. I remember work (on HCMV and human EC) by Jay Nelsons group saying that EC come in many different flavours. So, LSEC differ from lymphatic EC and EC from large vessels can differ from EC in the ramified capillary system of the lungs. So, have the authors secured that the findings and conclusions by Mouta Carreira remained undisputed in the EC community since 2001? The authors might feel more comfortable by convincing themselves that their interpretations of the data are based on firm facts in EC biology and that these mice were used for defining lymphatic EC also by others since 2001!

(5) Please explain also in the body of the text why tamoxifen needed to be used. This may not be immediately clear to every reader.

(6) p.8, 2nd§: Hematopoietic stem cells do certainly not derive from EC! What Zovein et al., 2008, show (again it should be checked if this is still current opinion in a moving field) is that they share the same endothelial progeny, which is not precisely the same. I doubt that mature EC do this.

Somehow related to this issue, Seckert et al. (JVI 2009) used sex-mismatched bone marrow chimeras for establishing latency and found that donor-derived CD31+CD146+ EC are MHC-II+ but do not become latently infected, whereas latent viral genome localized to recipient-genotype CD31+CD146+ EC not expressing MHC-II. This may have relevance for the authors’ data on the reconstitution of mice with WT bone marrow, in particular as lymphatic EC (as noted in the discussion) express MHC II. So, it might be that lymphatic EC are derived from transplanted bone marrow, but other DC not. This should at least be discussed.

DISCUSSION

(1) p9, 2nd §. Regarding the number of peptide-MHC-I complexes needed to activate CD8 T cells, I really miss any discussion of viral immune evasion proteins that should critically reduce or even block (Pinto et al. 2006) the presentation of SIINFEKL by latently infected lymphatic (or other) EC. To explain why MI can occur with viruses not deleted in immune evasion proteins, the authors should consider and discuss the recent work by Griessl et al. (Front. Immunol., 2021), in particular as this paper is supportive for the authors’ conclusions!

FIGURES

(1) Have the authors shown how many IFNg-expressing cells after MI (that is of the 1.4%) show the KLRG1+CD62L- phenotype?

(2) Figure 4 is somewhat complicated and might be improved. In 4D I got it that the pic in the middle is for Tek-Cre- and on the right is Tek-Cre +, but the pics are not helpful when they look alike.

(3) To aid the reader, please define LSL in the legend to S1

(4) Please tell the statistical method used also in the Figure Legends:

(5) It is unclear why in Fig. S1B 3 primers are listed although the legend refers only to primers 2 and 3.

Reviewer #2: -The authors show that the ∆gL-LSL-SL8 virus shows a very low level of recombination in vitro, and posit this ‘leakage’ may be responsible for the ~0.5% of SIINFEKL-specific T cell in mice that didn’t express Cre. Firstly, it is admirable they performed this control. Can the authors provide any data where they phenotyped these SIINFEKL cells at ~d8 in comparison with mice infected with MCMV-SIINFEKL without the LSL? This gets at the more general question of whether the initial priming of SIINFEKL CD8T cells may be different in the various Cre mice, and how it may different in infected endothelial cells compared to more classic professional APC.

-I suspect that some reviewers may be concerned that the authors used a BAC-derived Smith strain of MCMV lacking a functional Mck2 protein mutant Smith BAC. This is not a big concern in my case given all of the excellent comparative controls included in this work.

Reviewer #3: The following minor issues need to be clarified:

1. The Acronyms BECs and LECs are mentioned in figures next to the graphs, but are not clearly explained in figure legends, nor elsewhere in the text, as far as I could find.

2. The Figure 4D illustrates the availability of pMHC on Endothelial cells (EC) and non-endotelial ones. The figure is shown as if there are no pMHC molecules on beta-2m floxed cells in absence of Cre. IMO the pMHC should be available on these cells if the mice do not express Cre.

3. Based on results in Figure 4, the authors conclude that Ag expression on EC is non-essential for MI, because even in mice where B2m/flox was crossed to Tek-Cre they still observe some MI. However, they also show some residual B2m expression on EC in their system. Therefore, an alternative scenario, not excluded by their data, may be that the EC are essential but their model did not allow a rigorous proof, because the inflationary peptides could be presented on some of the endothelial cells. This needs to be discussed more clearly.

PLOS authors have the option to publish the peer review history of their article (what does this mean?). If published, this will include your full peer review and any attached files.

Reviewer #1: No

Reviewer #2: No

Reviewer #3: No

Figure Files:

Data Requirements:

Reproducibility:

References:

---

## [Editor Report · Decision Letter 1]

21 Dec 2022

Dear Dr. Hill,

We are pleased to inform you that your manuscript 'Latent CMV Infection of Lymphatic Endothelial Cells is Sufficient to drive CD8 T cell memory inflation' has been provisionally accepted for publication in PLOS Pathogens.

Best regards,

Robert F. Kalejta

Academic Editor

PLOS Pathogens

Blossom Damania

Section Editor

PLOS Pathogens

Kasturi Haldar

Editor-in-Chief

PLOS Pathogens

orcid.org/0000-0001-5065-158X

Michael Malim

Editor-in-Chief

PLOS Pathogens

orcid.org/0000-0002-7699-2064
---

## [Editor Report · Acceptance letter]

18 Jan 2023

Dear Dr. Hill,

We are delighted to inform you that your manuscript, "Latent CMV Infection of Lymphatic Endothelial Cells is Sufficient to drive CD8 T cell memory inflation," has been formally accepted for publication in PLOS Pathogens.

Best regards,

Kasturi Haldar

Editor-in-Chief

PLOS Pathogens

orcid.org/0000-0001-5065-158X

Michael Malim

Editor-in-Chief

PLOS Pathogens

orcid.org/0000-0002-7699-2064